# Antibacterial and Antifungal Activity of Functionalized Cotton Fabric with Nanocomposite Based on Silver Nanoparticles and Carboxymethyl Chitosan

Carlos Alberto Arenas-Chávez [1], Luciana Maria de Hollanda [2], Arturo A. Arce-Esquivel [3], Aldo Alvarez-Risco [4], Shyla Del-Aguila-Arcentales [5], Jaime A. Yáñez [6,7,*] and Corina Vera-Gonzales [8,9]

1 Departamento Académico de Biología, Universidad Nacional de San Agustín de Arequipa, Arequipa 04000, Peru; carenasc@unsa.edu.pe
2 Centro Universitário Metrocamp, Wyden University, Campinas, São Paulo 13035-350, Brazil; luciana.hollanda@professores.unimetrocamp.edu.br
3 Department of Health and Kinesiology, School of Community and Rural Health, The University of Texas, Tyler, TX 75799, USA; aarce@uttyler.edu
4 Facultad de Ciencias Empresariales y Económicas, Carrera de Negocios Internacionales, Universidad de Lima, Lima 15023, Peru; aralvare@ulima.edu.pe
5 Escuela Nacional de Marina Mercante "Almirante Miguel Grau", Callao 07021, Peru; sdelaguila@enamm.edu.pe
6 Vicerrectorado de Investigación, Universidad Norbert Wiener, Lima 15046, Peru
7 Gerencia Corporativa de Asuntos Científicos y Regulatorios, Teoma Global, Lima 15073, Peru
8 Departamento Académico de Química, Universidad Nacional de San Agustín de Arequipa, Arequipa 04000, Peru; cverag@unsa.edu.pe
9 Laboratorio de Preparación, Caracterización e Identificación de Nanomateriales (LAPCI NANO), Universidad Nacional de San Agustín de Arequipa, Arequipa 04000, Peru
* Correspondence: jaime.yanez@uwiener.edu.pe

**Abstract:** Cotton is the most widely used natural fiber for textiles; however, the capacity of cotton fibers to absorb large amounts of moisture, retain oxygen, and have a high specific surface area makes them more prone to microbial contamination, becoming an appropriate medium for the growth of bacteria and fungi. In recent years, the incorporation of silver nanoparticles in textile products has been widely used due to their broad-spectrum antibacterial activity and low toxicity towards mammalian cells. The aim of the current study is to continue the assessment of our developed nanocomposite and evaluate the antibacterial and antifungal activity of the nanocomposite based on silver nanoparticles and carboxymethyl chitosan (AgNPs-CMC) against *Escherichia coli*, *Staphylococcus aureus*, and *Candida albicans*, evaluated by the well diffusion method. The antibacterial activity against *E. coli* and *S. aureus* was also evaluated by the qualitative method of inhibition zone and the quantitative method of colony counting. Likewise, the antifungal activity of the functionalized fabric against *Candida albicans* and *Aspergillus niger* was determined by the inhibition zone method and the antifungal activity method GBT 24346-2009, respectively. The functionalized fabric showed 100% antibacterial activity against *E. coli* and *S. aureus* and good antifungal activity against *C. albicans* and *A. niger*. Our results indicate that the functionalized fabric could be used in garments for hospital use to reduce nosocomial infections.

**Keywords:** nanocomposite; functionalized fabric; antimicrobial; antifungal; cotton

## 1. Introduction

Cotton is the most widely used natural fiber for textiles due to its softness, low price, easy mass production, and it is particularly suitable for manufacturing medical products, healthcare products, and hygiene [1–3]. However, cotton fibers absorb large amounts of moisture [4], retain oxygen, and have a high specific surface area [5]. This also makes them more prone to microbial contamination, becoming an appropriate medium for the

growth of bacteria and fungi [6]. In this sense, cotton fibers with antimicrobial properties have attracted considerable attention due to their potential application in several fields such as health and medicine [7–13]. The use of biocides such as triclosan [14], quaternary ammonium compounds [15], or organosilicons [16,17] has been reported. However, these antimicrobial agents often produce highly toxic or undesirable by-products [18–20]. In recent years, nanocarrier systems have been widely used in various fields including nutraceuticals [21–31], pharmaceuticals [32–35], which could affect biodisposition [36–60], lymphatic transport [61], ophthalmic drug delivery [62], and toxicity [63–67]. The incorporation of silver nanoparticles in textile products has been widely used due to their broad-spectrum antibacterial activity and low toxicity towards mammalian cells [68–75]. The antimicrobial properties of silver nanoparticles are size-dependent because silver nanoparticles of different sizes have different surface/volume ratios, producing different antibacterial efficiency during their interaction with microorganisms [76–81]. The instability of silver nanoparticles has been reported because a variation in the size of the nanoparticles when they are applied directly on textiles can cause agglomeration of the nanoparticles, resulting in a decrease in the antimicrobial effect [82–85].

It has been described that the use of stabilizing agents, such as natural polymers, to form nanocomposites significantly improves the stability of the nanoparticles [86,87]. Polymer-based inorganic nanocomposites combine organic, inorganic, and nanomaterial compounds' unique mechanical, optical, and electrical properties. The polymer chains of these nanocomposites can contain reactive groups. In combination with the inorganic antimicrobial agents, they have exceptional advantages such as exhibiting synergistic antimicrobial effects, improving the adhesion to the substrates, avoiding agglomeration, and improving the stability of silver nanoparticles inside the polymer matrix [5,88]. There is only one study that used carboxymethyl chitosan to pad cotton fabric and then soak it in silver nitrate and black rice extract [89]. However, carboxymethyl cellulose has been preferably used as a reductant agent [90]. Other studies report the use of various reductant and stabilizing agents for silver nanoparticles such as acacia gum [91], a bionic mussel-like material named polydopamine (PDA) [74], ethanolamine [92], and carrageenan [93]. Furthermore, recently, a tri-component nanoparticle of silver, copper, and zinc oxide has been developed using polymethylol compound (PMC) or functionalized polyethyleneimine (FPEI) polymers as both reductant and stabilizing agents [94]. Other approaches have been implemented to improve properties of nanocomposites such as enhancing thermoelectric performance by realigning Fermi level [95], development of polyaniline derivatives towards multistimulus responsiveness by plasma activation [96], plasma treatment toward electrically conductive and superhydrophobic cotton fibers using polypyrrole [97], and self-cleanable cotton fibers using silver carbamate and plasma activation [98].

The novelty of our manuscript is centered around the assessment of antibacterial and antifungal activity of the nanocomposite previously synthesized and characterized by our research group based on silver nanoparticles and carboxymethyl chitosan (AgNPs-CMC) [99]. The nanocomposite obtained from the complex $[Ag(NH_3)_2]+$ was synthesized under the same conditions as $AgNO_3$, but at a basic pH. UV-VIS spectrophotometry verified the plasmon formation of silver nanoparticles at 410 nm for both silver sources [99]. Our results by Dynamic Light Dispersion (DLS) for $AgNO_3$, showed a monodisperse distribution of the nanocomposite with an average hydrodynamic size of 166.7 nm [99]. Infrared spectroscopy measurements with Fourier Transform (FT-IR) showed the inhibition of the spectral bands at 879 and 723 $cm^{-1}$ indicating the presence of AgNPs in the nanocomposite AgNPs-CMC [99]. The results of scanning electron microscopy (SEM-STEM) showed that the silver nanoparticles in the nanocomposite were spherical in shape and of a size of 5 to 20 nm [99]. The aim of the current study is to continue the assessment of our developed nanocomposite and evaluate its antibacterial and antifungal activity against *E. coli*, *S. aureus,* and *C. albicans* was evaluated by the well diffusion method. The antibacterial activity against *E. coli* and *S. aureus* was also evaluated by the qualitative method of inhibition zone and the quantitative method of colony counting.

The materials and methods are presented in Section 2. Section 3 provides the outcomes and discussion. Conclusions are described in Section 4.

## 2. Materials and Methods

### 2.1. Reagents and Materials

The following reagents were used: potassium dihydrogen phosphate (pa $\geq$99%) (Merck Millipore, Saint Louis, MO, USA), sodium hydroxide (pa $\geq$ 99%) (Merck Millipore, Saint Louis, MO, USA), trypticase soy agar (TSA) (Liofilchem, Abruzzi, Italy), Trypticase soy broth (TSB) (Liofilchem, Abruzzi, Italy), *Escherichia coli* (ATCC 25922) (Merck Millipore, Saint Louis, MO, USA), *Staphylococcus aureus* (ATCC 25923) (Merck Millipore, Saint Louis, MO, USA). *Candida albicans* and *Aspergillus niger* were provided by the Microbiology Laboratory of the Universidad Nacional de San Agustin (UNSA). The nanocomposite material (AgNPs-CMC) was provided by the Laboratory of Preparation, Characterization and Identification (LAPCI_NANO) of the Universidad Nacional de San Agustin de Arequipa (UNSA).

### 2.2. Synthesis of Nanocomposite, Preparation and Functionalization of Cotton Fabric

We have previously described the synthesis of the silver nanoparticles and carboxymethyl chitosan (AgNPs-CMC) [99]. Briefly, the nanocomposite synthesis was performed using 20 mL of silver nitrate solution (1 mM) with the dropwise addition of 30 mL O-CMC (0.025%) with constant stirring (700 rpm) for 30 min at 90 °C. The fabric was washed with a non-ionic detergent (2.0 g/L concentration) at 90 °C with constant stirring (30 rpm) for 15 min. Then, it was rinsed twice with distilled water at 60 °C with constant stirring (30 rpm) for 10 min in an Eco Dyer. One portion was kept at this point as control fabric, and the other portion was ready for functionalization. The fabric was dried at room temperature for 24 h. The fabric was functionalized using the exhaustion method [100] in an Eco Dyer. For this, one gram of fabric was submerged in 20 mL of an AgNPs-CMC nanocomposite solution under the following conditions: 90 °C temperature, constant stirring (30 rpm), liquor ratio of 1:20, and for 15 min. Then, the fabric was rinsed twice with distilled water at 30 °C temperature, constant stirring (30 rpm), liquor ratio of 1:20 for 15 min using the Eco Dyer. Finally, the fabric was dried at 80 °C for 15 min.

### 2.3. Antibacterial Activity of the Nanocomposite

The antibacterial activity was evaluated by the standard well diffusion method. Inoculation of the bacteria *E. coli* (ATCC 25922) and *S. aureus* (ATCC 25923) were prepared at a concentration of $1.02 \times 10^3$ CFU/mL and $1.36 \times 10^4$ CFU/mL, respectively. Then, 20 μL of the inoculum was measured and plated uniformly on the surface of the Mueller Hinton agar (MH). Then, 3 wells of 7 mm in diameter were made, distributed equidistantly in the Petri dish, and 20 μL of the nanocomposite was placed (AgNPs-CMC). The plates were then incubated for 24 h at 37 °C; then, the inhibition zone was measured around the well using a vernier caliper [101].

### 2.4. Antibacterial Activity of Cotton Fabric Functionalized with the Nanocomposite

2.4.1. Inhibition Zone Method

The antibacterial activity was evaluated by the qualitative zone inhibition method. Initially, inoculation of the bacteria *E. coli* (ATCC 25922) and *S. aureus* (ATCC 25923) was prepared at a concentration of $1.02 \times 10^3$ CFU/mL and $1.36 \times 10^4$ CFU/mL, respectively. Then, 20 μL of the inoculum was measured and plated uniformly in Petri dishes with Mueller–Hinton agar (MH) and allowed to dry for 10 min. On the agar surface, disks (previously sterilized by UV) of the control and functionalized fabric of 0.8 cm diameter were placed. The plates were then incubated for 24 h at 37 °C; then, the inhibition zone was measured around the fabric using a vernier caliper [102].

2.4.2. Colony Counting Method

The antibacterial effect of the fabric functionalized with nanocomposite AgNPs-CMC was tested using the standard quantitative method ASTM E-2149-10 against *E. coli* and *S. aureus*. Then, a fabric sample, weighing $1000 \pm 0.001$ g, was cut into small pieces with a size of about $1 \times 1$ cm$^2$, which were sterilized by UV radiation for 30 min. Then, these fabric pieces were immersed in a 250 mL bottle with 50 mL of 0.3 mM dihydrogen phosphate buffer solution, pH = 7.2 containing 1.5 to $3.0 \times 10^5$ CFU/mL of bacteria. The bottle was shaken at 150 rpm in a shaker at 37 °C, for 1 h. From each incubated sample, 1 mL of solution was taken and diluted to $10^{-1}$, $10^{-2}$, and $10^{-3}$ and then seeded on an agar plate. All plates were incubated at 37 °C for 24 h, and the colonies formed were counted with the naked eye. The percentage of bacterial reduction was determined as follows:

$$\text{Bacterial reduction in CFU (\%)} = x = \frac{B - A}{B} \times 100$$

where:

A = Colony-forming units (CFU)/mL for the bottle at the end time, after one hour of contact;
B = Colony-forming units (CFU)/mL for the bottle at time zero, after one minute of contact.

*2.5. Antifungal Activity of the Nanocomposite*

The antifungal activity of the nanocomposite was evaluated by the standard well diffusion method. An inoculum of *C. albicans* (ATCC 10231) was prepared at a $1.06 \times 10^7$ CFU/mL concentration. Then, 20 μL of the inoculum was measured and plated uniformly on the potato dextrose agar (PDA) surface. Then, 3 wells of 7 mm diameter were made equidistantly distributed in the petri dish, where 20 μL of the nanocomposite (AgNPs-CMC) was placed. The plates were then incubated for 24 h at 37 °C, after which time the zone of inhibition was measured around the well using a vernier caliper [101].

*2.6. Antifungal Activity of Cotton Fabric Functionalized with the Nanocomposite*

Antifungal activity was evaluated by the qualitative zone inhibition method. An inoculum of *C. albicans* of $1.06 \times 10^7$ CFU/mL was initially prepared. It was seeded in a PDA medium and then proceeded in the same way as in the case of *E. coli* and *S. aureus* described above.

*2.7. Antifungal Activity of the Cotton Fabric Functionalized with the Nanocomposite against a Filamentous Fungus*

The antifungal activity of the control cotton fabric and the functionalized fabric was evaluated against an isolated strain of *A. niger* according to the standard method of antifungal activity GBT 24346-2009 [103].

*2.8. Statistical Analysis*

Statistical analysis was performed by Student's *t* test using STATA Statistical Software: Release 15 (StataCorp LLC, College Station, TX, USA). A value of $p < 0.05$ (*) was considered to be statistically significant. All the studies were performed in triplicates.

**3. Results and Discussion**

*3.1. Antibacterial Activity of the Nanocomposite*

Figure 1 shows the inhibition of antibacterial activity by the well diffusion method against *E. coli* and *S. aureus*. There is an average inhibition halo of 5 mm for *E. coli* and 8 mm for *S. aureus*. The control exhibited no inhibition halo for *E. coli* and *S. aureus*. These results show that the nanocomposite has excellent antibacterial activity against these two types of Gram-negative and Gram-positive bacteria, respectively.

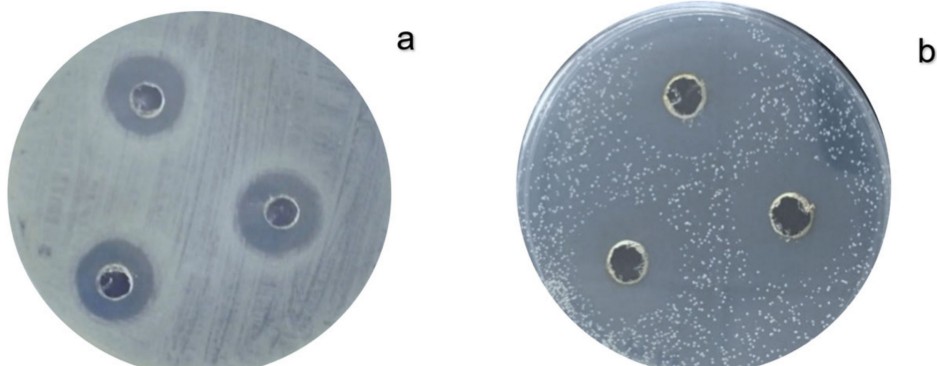

**Figure 1.** Antibacterial activity of the nanocomposite AgNPs-CMC against *E. coli* (**a**) and *S. aureus* (**b**).

*3.2. Antibacterial Activity of the Cotton Fabric Functionalized with the Nanocomposite*

The antibacterial activity of the fabric functionalized with the nanocomposite AgNPs-CMC was evaluated qualitatively by the zone of inhibition method and quantitatively by the colony counting method, according to the ASTM E-2149-10 technical standard. Figure 2 shows the antibacterial activity of the control and textile fabric functionalized with the nanocomposite. The textile functionalized with nanocomposite shows an average zone of inhibition of 1.5 mm and 2 mm around the textile for *E. coli* and *S. aureus*, respectively, which indicates that the textile functionalized with the nanocomposite possesses an antibacterial activity due to the action of the nanocomposite on the bacterial cells. The results in Table 1 and Figure 3 showed a bacterial reduction of 0% in the control fabric against *E. coli* and *S. aureus*. In contrast, the functionalized fabric exhibited a 100% bacterial reduction in both bacterial strains, showing the excellent antibacterial activity of the functionalized fabric compared to the control.

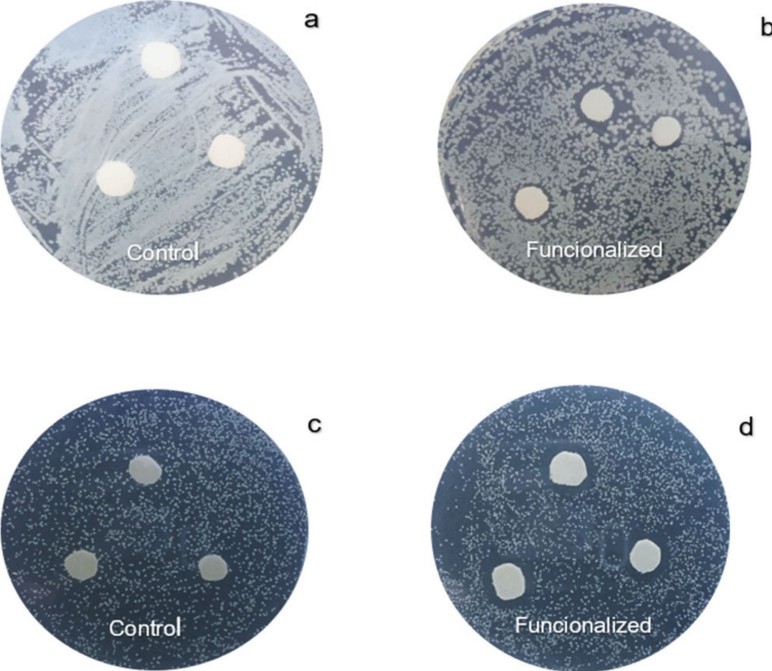

**Figure 2.** Antibacterial activity of the control and functionalized fabric against *E. coli* (**a**,**b**) and *S. aureus* (**c**,**d**).

**Table 1.** Antibacterial reduction of control fabric and fabric functionalized with nanocomposite AgNPs-CMC. Results expressed as mean $\pm$ standard deviation ($n = 3$).

| Sample | Antibacterial Activity | | | |
|---|---|---|---|---|
| | *E. coli* | | *S. aureus* | |
| | Bacteria UFC/mL | % reduction | Bacteria UFC/mL | % reduction |
| Control | $107500 \pm 13527$ | No reduction | $73167 \pm 2631$ | No reduction |
| Functionalized | $0 \pm 0$ * | 100% | $0 \pm 0$ * | 100% |

* $p < 0.05$.

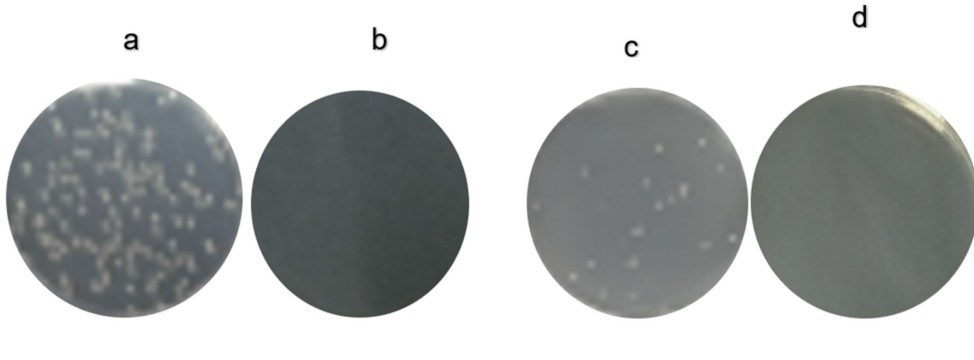

**Figure 3.** Antibacterial activity of the control and functionalized fabric against *E. coli* (**a**,**b**) and *S. aureus* (**c**,**d**).

Our antibacterial activity results are superior to those reported previously, with 80–90% bacterial reduction [57,68,93,94]. Similarly, these results for bacterial reduction correlate with Paszkiewicz et al. [104], where they used bimetallic silver and copper nanoparticles. The bactericidal mechanism of the silver nanoparticles is only partially known, and the antibacterial effect reported for the functionalized fabric could be explained based on the following mechanisms: the Ag+ ions formed from the oxidation of zero-valent silver (Ag°) interact with the sulfur of the proteins present in the bacterial cell membrane or intracellularly, which affects the viability of the bacterial cell. It has also been proposed that the silver/silver ion (AgNPs/Ag+) nanoparticles can act with the molecules of phosphorus present in DNA, producing an inactivation of DNA replication [105–109].

The release of Ag+ from the AgNPs can also catalyze the production of oxygen radicals that oxidize the molecular structure of the bacteria. This mechanism does not require direct contact between the antimicrobial agent Ag+ and the bacterium because the active oxygen produced diffuses from the textile to the surrounding environment [105,109].

*3.3. Antifungal Activity of the Nanocomposite*

Figure 4 shows the inhibition of antifungal activity by the well diffusion method against *C. albicans*. An average inhibition halo of 1.5 mm was observed exhibiting that the nanocomposite has antifungal activity against *C. albicans*. The control exhibited no inhibition halo for *C. albicans*.

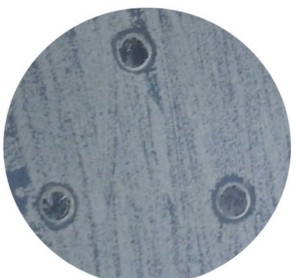

**Figure 4.** Antifungal activity of nanocomposite AgNPs-CMC against *C. albicans.*

*3.4. Antifungal Activity of the Cotton Fabric Functionalized with the Nanocomposite*

Figure 5 shows the antifungal activity of the control and textile fabric functionalized with the nanocomposite. The functionalized fabric shows an average zone of inhibition of 2 mm against *C. albicans*, which indicates that the fabric functionalized with the nanocomposite possesses antifungal activity. The antifungal activity of the control cotton and the functionalized fabric was evaluated against an isolated strain of *Aspergillus niger* according to the standard method GBT 24346-2009 [103,110]. Figure 6 shows the results of the antifungal activity of the control cotton and functionalized fabric with the nanocomposite.

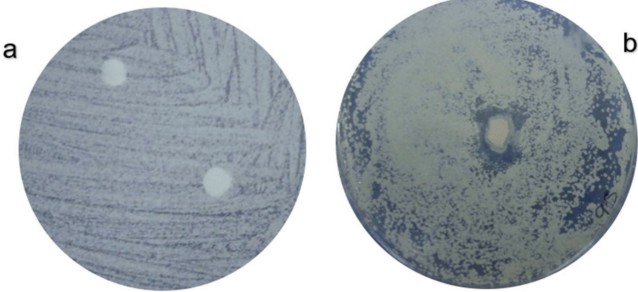

**Figure 5.** Antifungal activity of control (**a**) and functionalized (**b**) fabric against *C. albicans.*

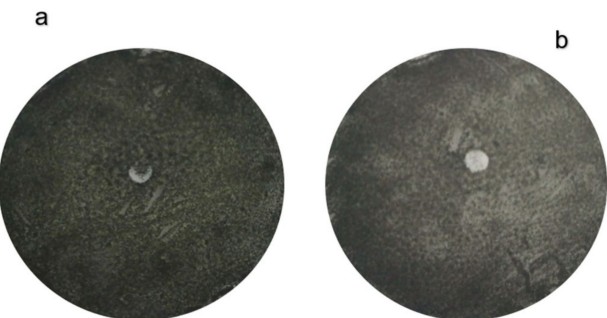

**Figure 6.** Antifungal activity of control (**a**) and functionalized (**b**) fabric against *A. niger.*

The control fabric (Figure 6a) is covered by the filamentous fungus, indicating that this fabric does not show resistance to the growth of *A. niger*. However, the textile functionalized with the nanocomposite (Figure 6b) presents a greater surface area without the growth of the fungus.

Gao et al. reported similar results, with the difference that they worked with a functionalized fabric with a nanocomposite based on P(DMDAAC-AGE)/Ag/ZnO, which presented an ability to inhibit the growth of *A. flavus* up to 5 days [103]. However, our functionalized textile showed a growth inhibition for up to 7 days, which shows a better antifungal effect.

## 4. Conclusions

Cotton is the most widely used natural fiber for textiles, with the recent incorporation of silver nanoparticles due to its broad-spectrum antibacterial activity and low toxicity towards mammalian cells. This work reported 100% antibacterial activity against *E. coli* and *S. aureus* and good antifungal activity against *C. albicans* and *A. niger* of our functionalized fabric with the nanocomposite based on silver nanoparticles and carboxymethyl chitosan (AgNPs-CMC). This functionalized fabric showed that our fabric could be used in garments for hospital use to reduce nosocomial infections, which invites further investigation and assessment of other applications in a larger number of microorganisms involved in nosocomial infections.

**Author Contributions:** Conceptualization, C.A.A.-C., L.M.d.H. and C.V.-G.; methodology, C.A.A.-C., L.M.d.H. and C.V.-G.; software, C.A.A.-C., L.M.d.H. and C.V.-G.; validation, C.A.A.-C., L.M.d.H. and C.V.-G.; formal analysis, C.A.A.-C., L.M.d.H. and C.V.-G.; investigation, C.A.A.-C., L.M.d.H. and C.V.-G.; resources, C.A.A.-C., L.M.d.H. and C.V.-G.; data curation, C.A.A.-C., L.M.d.H. and C.V.-G.; writing—original draft preparation, C.A.A.-C., L.M.d.H., C.V.-G., A.A.-R., S.D.-A.-A. and J.A.Y.; writing—review and editing, C.A.A.-C., L.M.d.H., C.V.-G., A.A.A.-E., A.A.-R., S.D.-A.-A. and J.A.Y.; visualization, C.A.A.-C., L.M.d.H., C.V.-G., A.A.A.-E., A.A.-R., S.D.-A.-A. and J.A.Y. All authors have read and agreed to the published version of the manuscript.

**Funding:** This research received no external funding.

**Data Availability Statement:** The data presented in this study are available on request from the corresponding author

**Conflicts of Interest:** The authors declare no conflict of interest.

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
