# Peer review of "Antibacterial and Antifungal Activity of Functionalized Cotton Fabric with Nanocomposite Based on Silver Nanoparticles and Carboxymethyl Chitosan"

_processes, doi:10.3390/pr10061088_

Round 1

Reviewer 1 Report

In “Antibacterial and anitifungal activity of functionalized cotton fabric with nanocomposite based on silver nanoparticles and carboxymethyl chitosan” by C. A. Areans-Chávez, L. M. Hollanda, A. Arce-Esquivel, A. Alvarez-Risco, S. Del-Aguila-Arecentales, J. A. Yañez, C. Vera-Gonzales submitted as an article to Processes, the authors describe simple antimicrobial and antifungal activity studies of previously reported nanocomposite NPsAg-CMQ and cotton fabic functionalized with nanocomposite NPsAg-CMQ. These studies are very basic, some without proper control data reported, presented without proper statistical analysis which makes it difficult to evaluate if the results from the studies are valid. I recommend this paper is reconsidered after major revisions to address the comments and concerns below.  

Comments and Concerns:

  1. General: All figure images lack scale bars
  2. General: All average measurements should be reported with their errors (standard deviation) reported.
  3. General: Are all the antimicrobial effects of these materials present statistically significantly different than their appropriate controls?
  4. General: What number of measurements were included in the zone of inhibition and CFU experiments? Are these replicates biological or technical?
  5. Page 4, Section 3.1: The antibacterial properties of NPsAg-CMQ should be reported in the context of a control (vehicle).
  6. Page 6, Section 3.3: The antifungal properties of NPsAg-CMQ should be reported in the context of a control (vehicle).
  7. General: Why were the antifungal properties of NPsAg-CMQ not tested against A. niger?
  8. General: are these materials bactericidal or bacteriostatic?
  9. General: Additional microbes are also involved in nosocomial infections, the impact of this study would be increased if the antimicrobial properties of these materials were evaluated against a larger number of microbes.

Reviewer 2 Report

The authors investigated the antimicrobial cotton fabrics that finished with silver nanoparticles and carboxymethyl chitosan. There are many items need to be considered before acceptance.

What is the novelty of this work?

Silver nanoparticles were purchased as mentioned by the authors. However, it is necessary to characterize silver nanoparticles at least by TEM.

Introduction section should be completely checked using these relevant references:

Bioactive Wound Dressing Gauze Loaded with Silver Nanoparticles Mediated by Acacia Gum

Bactericidal finishing of loomstate, scoured and bleached cotton fibres via sustainable in-situ synthesis of silver nanoparticles Development of multifunctional modified cotton fabric with tri-component nanoparticles of silver, copper and zinc oxideAntibacterial carrageenan/cellulose-nanocrystal system loaded with silver nanoparticles, prepared via solid-state technique

what is the role of carboxymethyl chitosan?

The treatment of cotton fabric with silver nanoparticle should be added in details.

The treated cotton fabrics should be characterized in terms of surface morphology and elemental analysis (SEM and EDX).

Reviewer 3 Report

title and abstract must be revised to reflect the significance of research.

novelty is unclear?

SEM study of fabrics should be included.

TEM and XRD or AgNPs should be studied.

many references are missed and must be cited in the introduction section;

Journal of Colloid and Interface Science 582 (2021): 112-123.

Chemical Engineering Journal 409 (2021): 128291.

Cellulose 27, no. 5 (2020): 2913-2926.

Reactive and Functional Polymers 159 (2021): 104810.

Cellulose 28, no. 2 (2021): 1105-1121.

Round 2

Reviewer 1 Report

1.  Okay

2. Okay

3. Okay 

4. Okay

5. Okay

6. Okay 

7. Okay

8. Okay

9. Okay 

Reviewer 2 Report

Accept